



# Brief communication: A double Gaussian wake model

Johannes Schreiber, Amr Balbaa, and Carlo L. Bottasso

Wind Energy Institute, Technische Universität München, 85748 Garching bei München, Germany

**Correspondence:** Carlo L. Bottasso (carlo.bottasso@tum.de)

**Abstract.** In this paper, an analytical wake model with a double Gaussian velocity distribution is presented, improving on a similar formulation by Keane et al. The choice of a double Gaussian shape function is motivated by the behavior of the near wake region, observed in numerical simulations and experimental measurements. The method is based on the conservation of momentum principle, while stream-tube theory is used to determine the wake expansion at the tube outlet. The model is calibrated and validated using large eddy simulations replicating scaled wind turbine experiments. Results show that the tuned double Gaussian model is superior to a single Gaussian formulation in the near wake region.

## 1 Introduction

Analytical engineering wind farm models are low-fidelity approximations used to simulate the performance of wind power systems. A wind farm model includes both a model of the wind turbines and a model of the modifications to the ambient flow induced by their wakes, together with their mutual interactions. Analytical wake models, as opposed to high-fidelity computational fluid dynamics (CFD) models, are simple, easy to implement and computationally inexpensive. In fact, they only simulate macroscopic average effects of wakes and not their small scales and turbulent fluctuations. Engineering wake models find applicability in all those cases that do not need to resolve small spatial and fast temporal scales, such as the calculation of the power production of a wind plant over a sufficiently long time horizon. Such models are also extremely useful in optimization problems, where a large number of simulations might be required before a solution is reached, or where calculations need to be performed on the fly in real-time. Analytical wake models are thus often utilized in wind farm layout planning and in the emerging field of wind farm and wake control (Scholbrock, 2011; Churchfield, 2013; Boersma et al., 2017).

Because of their indisputable usefulness, engineering wake models have been extensively studied in the literature. The Jensen (PARK) formulation is one of the most widely used wake models, to the extent that it is sometimes considered as the industry standard (Keane et al., 2016). The model was first introduced by Jensen (1983), and later further developed by Katic et al. (1986). Other widely used and cited wake models include the Frandsen model (Frandsen et al., 2006), the FLORIS model (Gebraad et al., 2014), and the EPFL Gaussian models (Bastankhah and Porté-Agel, 2014, 2016).

All such models have been designed to faithfully represent the average flow properties of the far wake region. However, in the near wake (which is usually defined as the region up to about 4 diameters (D) downstream of the rotor disk), the models seem to lack accuracy. Nowadays, onshore wind farms tend to be closely packed, and turbine spacing often reaches values





below 3D. This raises the necessity of developing models that accurately represent the wake not only far away from the rotor disk, but also in the near and mid-wake regions.

Keane et al. (2016) developed a wake model featuring a double Gaussian velocity deficit distribution, in an attempt to formulate a model that closely resembles observed speed distributions in both the near and far wake regions. In fact, while a
single Gaussian function is considered to be a good approximation of the wake velocity distribution in the far wake (Bastankhah and Porté-Agel, 2014, 2016), the near wake is better approximated using a double Gaussian distribution. This is due to the presence of two peaks in the speed profiles close to the rotor disk, as also observed in experimental measurements and high-fidelity CFD simulations (Wang et al., 2017). The double Gaussian model by Keane et al. (2016), which is referred to as the Keane model in this paper, was developed in a similar fashion to the EPFL Gaussian model (Bastankhah and Porté-Agel, 2014),
and was intended to respect the principles of mass and momentum conservation.

In this short note, a double Gaussian wake model, based on Keane's model and with emphasis on near wake flow behavior, is derived, calibrated and validated. The present formulation addresses and resolves some issues found in the original implementation of Keane et al. (2016), primarily concerning momentum conservation. In addition, the wake expansion function is defined such that mass flow deficit conservation is achieved at the stream-tube outlet.

This paper is organized as follows. The derivation of the double Gaussian wake model is detailed in Section 2, along with the formulation of the wake expansion function. In Section 3, the model is tuned and validated, using both experimental measurements obtained with scaled models in a boundary layer wind tunnel and by numerical results of high-fidelity large eddy simulations (LES). Additionally, the performance of the double Gaussian model is compared to a standard single Gaussian formulation. Finally, concluding remarks and future work recommendations are given in Section 4. Appendix A derives some
integrals appearing in the formulation.

## 2 Wake model description

### 2.1 Double Gaussian velocity deficit

The double Gaussian wake model is derived in a similar way to the Frandsen (Frandsen et al., 2006) and EPFL single Gaussian models (Bastankhah and Porté-Agel, 2014). Following their approach, the conservation of momentum principle is applied
on an ansatz velocity deficit distribution, which includes an amplitude function. Thereby, an expression for the amplitude is obtained that assures conservation of momentum.

At the downstream distance $x$ from the wind turbine rotor and at the radial distance $r$ from the wake centerline, the wake velocity deficit $U_\infty - U(x,r)$ is modeled as the product of the normalized double Gaussian function $g(r,\sigma(x))$, which dictates the spatial shape of the deficit, with the amplitude function $C(\sigma(x))$. This yields

$$\frac{U_\infty - U(x,r)}{U_\infty} = C(\sigma(x))g(r,\sigma(x)), \tag{1}$$





where $U_\infty$ represents the ambient wind speed and $U(x,r)$ the local flow velocity in the wake. The double Gaussian wake shape function, which is symmetric with respect to the wake center, is defined as

$$g(r,\sigma(x)) = \frac{1}{2}\left(e^{D_+} + e^{D_-}\right), \qquad D_\pm = \frac{-(r \pm r_0)^2}{2\sigma^2(x)}, \tag{2}$$

where $r_0$ is the radial position of the Gaussian extrema. The standard deviation of the Gaussian function, noted $\sigma(x)$, represents the width (cross-section) of each of the two single Gaussian profiles. The wake expands with downstream distance $x$, causing the transformation of the initial double Gaussian profile in the near wake, through a flat-peak transition region, into a nearly single Gaussian profile in the far wake. The wake expansion function is discussed in further detail in §2.2.

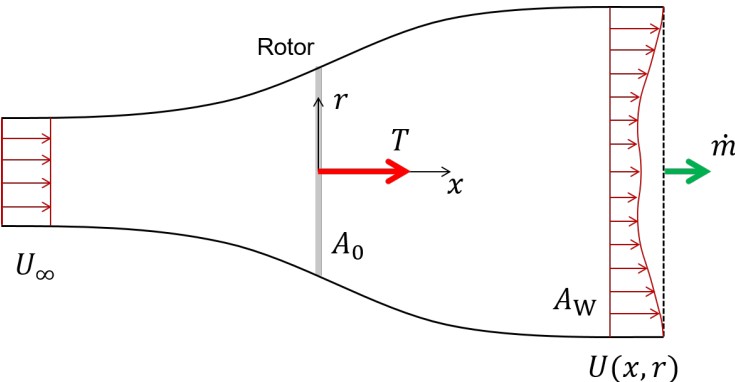

**Figure 1.** Stream tube with nomenclature: $U_\infty$ is the ambient wind speed; $U(x,r)$ is the local flow velocity in the wake at the downstream position $x$ and radial distance $r$ from the wake centerline; $\dot{m}$ is the mass flow rate through the stream tube; $A_W$ is a planar cross-sectional area large enough to contain the wake deficit, and $A_0$ is the rotor disk area; $T$ is the thrust force (by the principle of action and reaction, an equal and opposite force is applied by the rotor onto the flow).

The conservation of momentum principle is now applied on the ansatz velocity deficit distribution, using the amplitude function $C(\sigma(x))$ as a degree of freedom. Accordingly, the axial thrust force $T$ is related to the rate of change of momentum $p$ of the flow throughout the stream tube (see Fig. 1), i.e.

$$T = \frac{\mathrm{d}p}{\mathrm{d}t} = \dot{m}\Delta\tilde{U} = \rho \int_{A_W} U(x,r)\left(U_\infty - U(x,r)\right)\mathrm{d}A_W, \tag{3}$$

where $\dot{m}$ is the mass flow rate through the stream tube, $\Delta\tilde{U}$ an effective wake velocity deficit, $\rho$ the air density and $A_W$ a planar cross-section at least large enough to contain the wake deficit. Equation (3) is only valid if there is an equal pressure and negligible flow acceleration at the inlet and outlet sections of the the stream tube and, additionally, if shear forces on the control volume can be neglected. The thrust force $T$ is customarily expressed through the non-dimensional thrust coefficient $C_T$ as

$$T = \frac{1}{2}\rho A_0 U_\infty^2 C_T, \tag{4}$$





where $A_0$ is the rotor swept area.

If the wake velocity, defined in Eqs. (1) and (2), is substituted into the Eq. (3), one obtains

$$T = \rho\pi U_\infty^2 C(\sigma) \int\limits_0^\infty \left( e^{D_+} + e^{D_-} - \frac{C(\sigma)}{2}\left(e^{2D_+} + e^{2D_-} + 2e^{D_+ + D_-}\right)\right) r\,dr. \tag{5}$$

Note that, as the double Gaussian wake expands all the way to infinity, the integral boundary is set accordingly. The integration of Eq. (5), whose details are provided in Appendix A, yields

$$T = \rho\pi U_\infty^2 C(\sigma)\left(M - C(\sigma)N\right), \tag{6}$$

where

$$M = 2\sigma^2 e^{\frac{-r_0^2}{2\sigma^2}} + \sqrt{2\pi}r_0\sigma\,\mathrm{erf}\left(\frac{r_0}{\sqrt{2}\sigma}\right), \tag{7a}$$

$$N = \sigma^2 e^{\frac{-r_0^2}{\sigma^2}} + \frac{\sqrt{\pi}}{2}r_0\sigma\,\mathrm{erf}\left(\frac{r_0}{\sigma}\right). \tag{7b}$$

By substituting the thrust given by Eq. (4) into Eq. (6), and solving the resulting quadratic equation for the amplitude function $C(\sigma)$, one obtains

$$C_\pm(\sigma(x)) = \frac{M \pm \sqrt{M^2 - \frac{1}{2}NC_\mathrm{T}d_0^2}}{2N}, \tag{8}$$

where $d_0 = \sqrt{4A_0/\pi}$ is the rotor diameter. Both solutions of the amplitude function $C(\sigma)$ would theoretically lead to the conservation of momentum at all downstream distances. However, the velocity profiles obtained by using $C_+(\sigma)$ are characterized by a negative speed (i.e., in the direction opposite to the ambient flow), and thus $C_+(\sigma)$ is deemed to be a nonphysical solution. Therefore, the true solution for the amplitude function is $C_-(\sigma)$. In addition, a momentum-conserving solution exists only if $M^2 - 1/2\,NC_\mathrm{T}d_0^2 \geq 0$, which might not always be the case for large values of $C_\mathrm{T}$.

The derived expressions for $M$ and $N$ presented in this paper differ from the results reported in the original publication by Keane et al. (2016), even though all assumptions are identical. The expressions reported in the original paper were also evaluated numerically, yielding nonphysical results that violate the conservation of mass and momentum underlying the formulation.

## 2.2 Wake expansion function

In the previous section, following the conservation of momentum, the shape of the double Gaussian wake deficit has been defined as a function of the Gaussian parameter $\sigma$. In this section, a wake expansion function $\sigma(x)$ is introduced, which is linear with respect to the downstream distance $x$. By mass conservation, the wake expansion at the position of the stream tube outlet is therefore identified.

In previous work by Frandsen et al. (2006) and Bastankhah and Porté-Agel (2014), stream tube theory was employed to derive an equation for the initial wake width at the turbine rotor. Thereby, the number of tunable parameters of the wake



expansion function is reduced, facilitating model calibration. However, this approach includes the assumption that the stream tube outlet is located exactly at the turbine rotor itself, which is hardly true. Results indicate that the derived initial wake width is too large to fit experimental measurements, which in turn requires a model re-tuning (Bastankhah and Porté-Agel, 2014).

In the present work, the stream tube outlet is not assumed to be located at the turbine rotor ($x = 0$), but at the unknown
downstream position $x_0$. Therefore, the expansion function is defined as

$$\sigma(x) = k^*(x - x_0) + \epsilon, \tag{9}$$

where parameter $k^*$ controls the rate of expansion, while $\epsilon$ represents the wake expansion at $x_0$. The wake expansion function is assumed to be linear as in Bastankhah and Porté-Agel (2014).

To derive $\epsilon$, mass conservation between the Betz stream tube and the wake model is enforced. Starting from Eq. (3), Frandsen
et al. (2006) and Bastankhah and Porté-Agel (2014) show that the mass flow deficit rate at the outlet of a Betz stream tube (noted ST) can be written as

$$\dot{m}_{\mathrm{ST}} = \rho \int_{A_{\mathrm{ST}}} \frac{U_\infty - U_{\mathrm{ST}}}{U_\infty} \, \mathrm{d}A_{\mathrm{ST}} = \rho \frac{\pi}{8} d_0^2 \beta \left( 1 - \sqrt{1 - \frac{2}{\beta} C_{\mathrm{T}}} \right), \tag{10}$$

where $U_{\mathrm{ST}}$ is the uniform cross-sectional wake velocity. In this expression, $\beta$ is the ratio between the stream tube outlet area $A_{\mathrm{ST}}$ and the rotor disk area $A_0$, which can be expressed as a function of the thrust coefficient $C_{\mathrm{T}}$ as

$$\beta = \frac{A_{\mathrm{ST}}}{A_0} = \frac{1}{2} \frac{1 + \sqrt{1 - C_{\mathrm{T}}}}{\sqrt{1 - C_{\mathrm{T}}}}. \tag{11}$$

At the Betz stream tube outlet ($x = x_0$), the mass flow deficit rate of the double Gaussian (noted DG) wake model writes

$$\dot{m}_{\mathrm{DG}} = \rho \int_{A_{\mathrm{W}}} \frac{U_\infty - U(x_0, r)}{U_\infty} \, \mathrm{d}A_{\mathrm{W}} = \rho \pi M(\epsilon) \frac{M(\epsilon) - \sqrt{M(\epsilon)^2 - \frac{1}{2} N(\epsilon) C_{\mathrm{T}} d_0^2}}{2N(\epsilon)}. \tag{12}$$

By equating both mass flow deficits (i.e., $\dot{m}_{\mathrm{DG}} = \dot{m}_{\mathrm{ST}}$), the initial wake expansion $\epsilon$ can be derived. The solution was computed numerically as a function of the thrust coefficient $C_{\mathrm{T}}$ and the spanwise location of the Gaussian extrema $r_0$. The resulting
surface is presented in Fig. 2. Note that the solution of stream tube theory is defined only in the range $0 \leq C_{\mathrm{T}} < 1$, and that $\epsilon$ tends to infinity as the thrust coefficient approaches the value of 1, due to mass conservation.

The remaining parameters $x_0$ and $k^*$ in the linear wake expansion function expressed by Eq. (9) are not explicitly modeled, and should be tuned based on experimental measurements or high-fidelity simulations, as shown in the next section. Note that the underlying momentum conservation statement expressed by Eq. (3) has only been defined for ambient pressure. Therefore,
the formulation is, strictly speaking, only valid for $x \geq x_0$. However, as pressure recovers rapidly immediately downstream of the rotor, resonable approximations can also be expected for $x < x_0$. Finally, $k^*$ is expected to depend on atmospheric conditions (Peña et al., 2016) and turbine thrust (Bottasso et al., 2019).

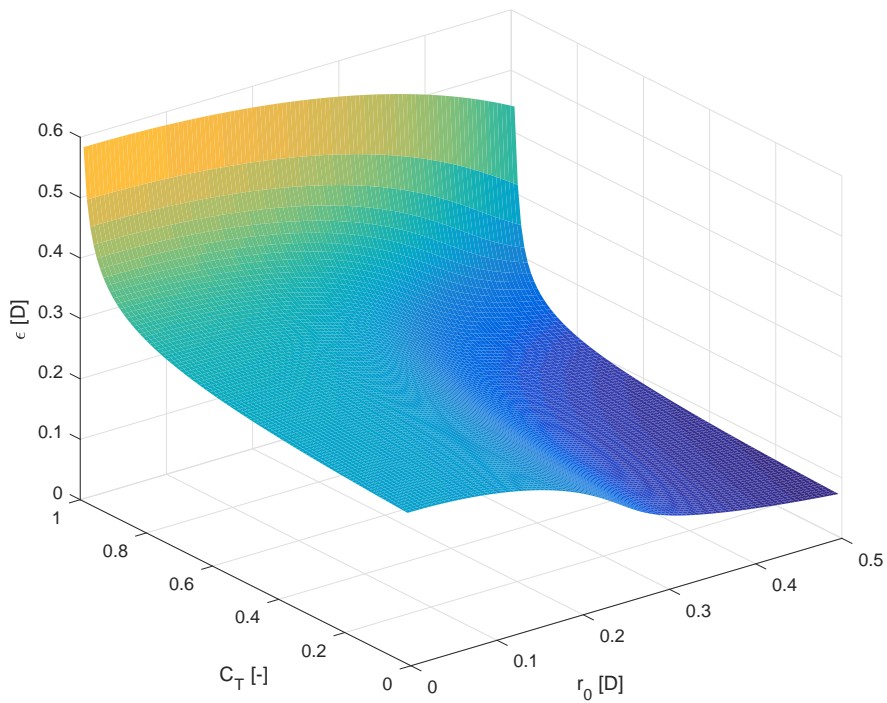

**Figure 2.** Visualization of the width of the double Gaussian function $\epsilon$ at the stream tube outlet, as a function of the thrust coefficient $C_T$ and the position of the Gaussian extrema $r_0$.

## 3 Model calibration and validation

### 3.1 Experimental and simulation setup

130    To calibrate and validate the double Gaussian wake model, time-averaged flow measurements from an LES numerical solution have been used. The CFD simulation replicates an experiment conducted with the scaled `G1` wind turbine (Campagnolo et al., 2017; Bottasso et al., 2019), which has a 1.1 m rotor diameter and a 0.8 m hub height. Its design operating tip speed ratio is 8 and its rated rotor speed is 850 rpm. The `G1` model is designed such that the characteristics of its wake are realistic in terms of shape, velocity deficit and recovery. In addition, the model features closed-loop pitch, torque and yaw control, as well as

135    load sensors located at the shaft and tower base (Campagnolo et al., 2017). The experiment was conducted with a single `G1` wind turbine model in the 36 m × 16.7 m × 3.84 m boundary layer wind tunnel at Politecnico di Milano. The wake profile was measured using hot-wire probes at the downstream distances $x/\mathrm{D} = \{1.4, 1.7, 2, 3, 4, 6, 9\}$ from the turbine. At each downstream location, the velocity was measured at hub height at different lateral positions $y$, and then time-averaged to obtain a steady-state value. The ambient wind velocity within the wind tunnel was measured using a pitot tube placed upwind of



the `G1` model. The wind tunnel experiment was conducted with a 5 ms$^{-1}$ ambient wind speed and a turbulence intensity of approximately 5%, with the wind turbine operating at $C_\mathrm{T} \approx 0.75$.

A complete digital copy of the experiment was developed with the LES simulation framework developed by Wang et al. (2017), which includes the passive generation of a sheared and turbulent flow, an actuator line model of the wind turbine implemented with the FAST aeroveroelastic simulator (Jonkman et al., 2005), as well as the tunnel walls. The simulation model includes also a slight lateral non-uniformly of the inflow, in the form of a 2.7% horizontal shear, caused by the wind tunnel internal layout upstream of the test chamber and by the tunnel fans. The proposed double Gaussian wake model was calibrated and validated using time-averaged CFD simulation results at the same downstream distances as the experiments, numerical and experimental measurements being in excellent agreement with each other.

### 3.2 Parameter identification and results

The double Gaussian model proposed in this work has three tunable parameters: $k^*$, $x_0$, and $k_\mathrm{r}$. Parameters $k^*$ and $x_0$ are used to describe the wake expansion downstream of the turbine rotor, as expressed by Eq. (9). The third parameter, $k_\mathrm{r}$, is defined as $r_0 = k_\mathrm{r}/2$ and describes the position of the Gaussian extrema. When $k_\mathrm{r} = 1$ the curve extrema are located at the tip of the rotor blades, while for $k_\mathrm{r} = 0$ the two Gaussian functions coincide at the wake center, leading to a single Gaussian wake profile.

In the original formulation by Keane, $k_\mathrm{r}$ was fixed at 75% blade-span, as it was argued that most lift is extracted from the flow at this location. In the present work the parameter is tuned based on measurements, as the assumed 75% blade-span position did not lead to satisfactory results.

The goal of model calibration is to ensure that the wind velocity profiles match the reference data set as closely as possible. To this end, the squared error between the wake model and CFD-computed wake profiles is minimized with respect to the free parameters. This estimation problem was solved using the Nelder-Mead simplex algorithm implemented in the MATLAB function `fminsearch` (Lagarias et al., 1998). To ensure the generality of the results, only a subset of the reference data was used for parameter estimation (namely, the downstream distances 1.7 D, 3 D and 6 D), while the others were used for verification purposes.

The identified parameters are presented in Table 1. The Gaussian extrema were found to be at approximately 53.5% of blade-span ($k_\mathrm{r} = 0.535$), while the wake width at $x_0$ is $\epsilon = 0.23$ D. Model calibration also resulted in the positioning of the stream tube outlet at $x_0 = 4.55$ D, which appears to be a realistic value for the investigated turbine.

**Table 1.** Identified model parameters.

| Operating conditions | | Parameters | | |
|---|---|---|---|---|
| $U_\infty$ [ms$^{-1}$] | $C_\mathrm{T}$ [-] | $k^*$ [-] | $x_0$ [D] | $k_\mathrm{r}$ [-] |
| 5.00 | 0.75 | 0.011 | 4.55 | 0.535 |



Figure 3 shows the experimental wake measurements using black circles, for all available downstream distances. The CFD results, shown using red × symbols, are almost identical to the experimental measurements, highlighting the quality of the LES simulations. The predictions of the double Gaussian model are shown in blue solid lines.

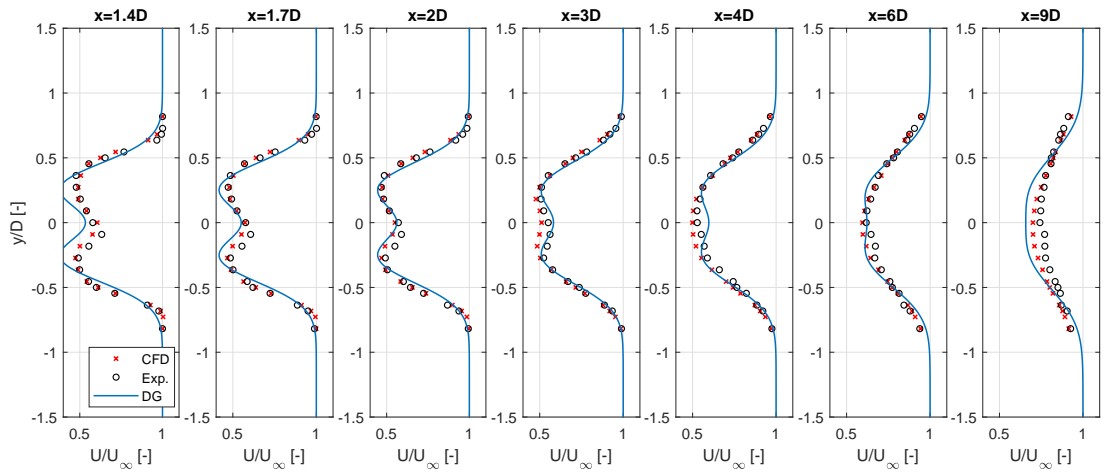

**Figure 3.** Normalized wind velocity profiles of the double Gaussian model (blue solid), compared to experimental measurements (black circles) and CFD simulations (red × symbols). The distances 1.7 D, 3 D and 6 D were used for model calibration.

The model exhibits good generality, as demonstrated by the good matching of the profiles at distances that were not used for
model estimation. Especially in the near wake region up to about 3 D, the placement of the Gaussian extrema appears to be in good agreement with the measured one.

The performance of the model clearly depends on the data used for its calibration. Using reference data close to the turbine rotor is important for accurately gauging the positions of the velocity profile extrema, while a wider rage of distances leads to an improved expansion behavior. In the present case, more data from the near wake region (1.7 D and 3 D) was considered in
the tuning process than in the far wake (6 D). This leads to a slight overestimation of the velocity deficit at 9 D, which could be attributed to an underestimation of the expansion slope $k^*$. However, tuning the model using the entire set of reference data points leads to only small differences in the identified parameters, which in turn produce wake profiles that are fairly similar to the ones presented here. On the other hand, identifying the model using only data points from the far wake resulted in better fitting results at 9 D, but with either very small values of $r_0$ —which lead to nearly single-Gaussian profiles—, or with high
values of the $k^*$ expansion slope —which lead to non-physical solutions of Eq. (8) for the amplitude function in the near wake region, due to excessively small Gaussian widths.

Figure 4 depicts with a solid blue line and $*$ symbols the root mean square error (RMSE) between the DG wake model (based on the parameters reported above, obtained from measurements at 1.7 D, 3 D and 6 D) and the reference CFD data as a function of downstream distance. To identify a lower error bound, the DG wake model parameters were also tuned separately at each
downstream distance, obtaining seven different local parameterizations. The corresponding RMSE with respect to the CFD




solution is reported in the same figure using a dashed blue line. The small difference between the two curves shows that the single (global) parameterization computed using only three distances is only marginally sub-optimal, in the sense that it is very close to the best possible fitting that a double Gaussian shape function can achieve. The plot shows also a slight increase of the difference between the two curves in the far wake region, which can again be attributed to the fact that only one large-distance 190 (6 D) measurement was used in the global tuning.

As a comparison, Fig. 4 also shows the results obtained with the EPFL single Gaussian (SG) model (Bastankhah and Porté-Agel, 2014). The SG model was identified with the same procedure and measurements used for the DG model, obtaining $\epsilon_{\text{SG}} = 0.3177$ and $k^*_{\text{SG}} = 0.0082$; the corresponding RMSE with respect to the CFD results is reported in the figure using a solid red line with circles. The lower error bound for the SG model, here again obtained by tuning the parameters independently 195 at each one of the seven available downstream distances, is shown using a dashed red line. As expected, the SG wake model shows a significantly larger RMSE in the near wake region than in the DG case. In fact, here the wake profile is indeed characterized by two peaks, so that the double Gaussian shape function allows for a more precise representation of the actual flow characteristics. Here again it should be noted that the difference between the global and local parameterizations is quite small, which strengthens the conclusion that improved results are due to the ansatz velocity deficit distribution and not to the 200 specific parameterizations.

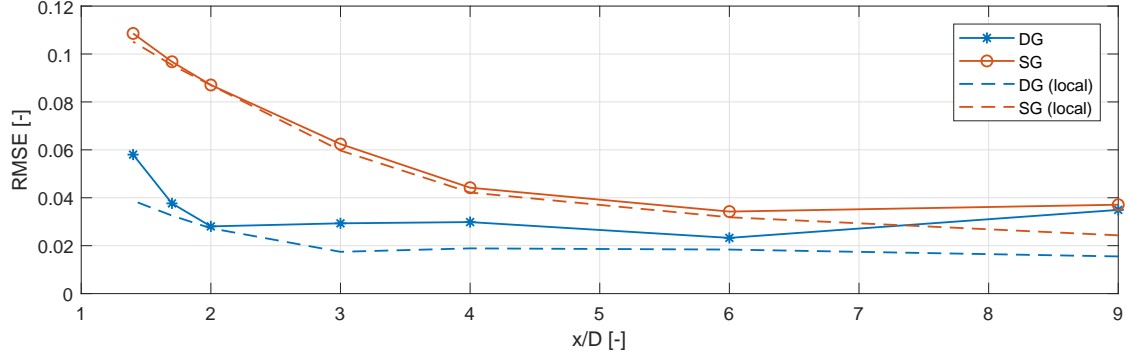

**Figure 4.** Root mean square error between the reference CFD velocity deficit data and the engineering wake models. Double Gaussian (DG) wake model identified using measurements at 1.7 D, 3 D and 6 D: solid blue line with ∗ symbols. DG wake model parameterized locally at each downstream distance: dashed blue line. Single Gaussian (SG) EPFL wake model identified using measurements at 1.7 D, 3 D and 6 D: solid red line with circles. SG EPFL wake model parameterized locally at each downstream distance: dashed red line.

## 4 Conclusions

This short paper presented an analytical double Gaussian wake model. The proposed formulation corrects and improves a previously published model proposed by Keane et al. The shape of the velocity deficit distribution in the wake is described by two Gaussian functions, which are symmetric with respect to the wake center, while the amplitude of the velocity deficit is





derived using the principle of momentum conservation. A linear expansion of the width of the Gaussian profiles was assumed, and stream tube theory was used to estimate the conditions at the stream tube outlet.

The model was calibrated and validated using a set of time-averaged CFD simulation results, which replicate wind tunnel experiments performed with a scaled wind turbine in a boundary layer wind tunnel. Results show that the model fits the reference data with good accuracy, especially in the near wake region where a single Gaussian wake is unable to describe the
typically observed bimodal velocity profiles. In the far wake, a slight overestimation of the wake deficit could be observed. It is speculated that this might be due to the wake expansion gradient being slightly different in the near and far wake regions. This claim, however, would need additional work to be substantiated. Future work could extend the wake model to include wake deflection, which could be done in a rather straightforward manner by following Bastankhah and Porté-Agel (2016). In this case, a non-symmetric double Gaussian shape function could be used to model the kidney shape of a deflected wake (Bartl
et al., 2018).

*Code and data availability.* A MATLAB implementation of the wake model and the data contained in this article can be obtained by contacting the authors.

**Appendix A: Integration of the momentum flux conservation formula**

Equation (5) can be written as

$$T = \rho \pi U_\infty^2 C(\sigma) \left( M - C(\sigma) N \right),$$ (A1)

where

$$M = \int_0^\infty \left( e^{D+} + e^{D-} \right) r \mathrm{d}r,$$ (A2a)

$$N = \int_0^\infty \left( \frac{1}{2} \left( e^{2D+} + e^{2D-} + 2e^{D++D-} \right) \right) r \mathrm{d}r.$$ (A2b)

In the following, integrals $M$ and $N$ are solved to obtain Eq. (7a) and (7b).

**A1 Derivation of $M$**

$M$ can be split into two terms:

$$M = I_1 + I_2.$$ (A3)

Term $I_1$ is defined as:

$$I_1 = \int_0^\infty e^{D+} r \mathrm{d}r = \lim_{R \to \infty} \int_0^R r e^{\frac{-(r+r_0)^2}{2\sigma^2}} \mathrm{d}r.$$ (A4)





Noticing that $D_\pm = \frac{-(r \pm r_0)^2}{2\sigma^2(x)}$, one gets

$$I_1 = \lim_{R \to \infty} \left[ -\sigma^2 e^{\frac{-(r+r_0)^2}{2\sigma^2}} - \frac{\sqrt{\pi} r_0 \sigma \, \mathrm{erf}\left( \frac{r+r_0}{\sqrt{2}\sigma} \right)}{\sqrt{2}} \right]_0^R , \tag{A5a}$$

$$= \sigma^2 e^{\frac{-r_0^2}{2\sigma^2}} - \frac{\sqrt{2\pi} r_0 \sigma}{2} \, \mathrm{erfc}\left( \frac{r_0}{\sqrt{2}\sigma} \right), \tag{A5b}$$

where erf is the Gauss error function

$$\mathrm{erf}(x) = \frac{1}{\sqrt{\pi}} \int_{-x}^{x} e^{-t^2} \, \mathrm{d}t, \tag{A6}$$

and $\mathrm{erfc}(x) = 1 - \mathrm{erf}(x)$ its complementary function. Similarly, $I_2$ writes:

$$I_2 = \int_0^\infty e^{D_-} r \, \mathrm{d}r = \lim_{R \to \infty} \int_0^R r e^{\frac{-(r-r_0)^2}{2\sigma^2}} \, \mathrm{d}r, \tag{A7}$$

and its integral is computed as

$$I_2 = \lim_{R \to \infty} \left[ -\sigma^2 e^{\frac{-(r-r_0)^2}{2\sigma^2}} + \frac{\sqrt{\pi} r_0 \sigma \, \mathrm{erf}\left( \frac{r-r_0}{\sqrt{2}\sigma} \right)}{\sqrt{2}} \right]_0^R , \tag{A8a}$$

$$= \sigma^2 e^{\frac{-r_0^2}{2\sigma^2}} + \frac{\sqrt{2\pi} r_0 \sigma}{2} \, \mathrm{erfc}\left( \frac{-r_0}{\sqrt{2}\sigma} \right). \tag{A8b}$$

Combining the previous results, one gets Eq. (7a), i.e.

$$M = I_1 + I_2 = 2\sigma^2 e^{\frac{-r_0^2}{2\sigma^2}} + \sqrt{2\pi} r_0 \sigma \, \mathrm{erf}\left( \frac{r_0}{\sqrt{2}\sigma} \right). \tag{A9}$$

**A2   Derivation of $N$**

Term $N$ can be split into three terms

$$N = \frac{1}{2} \left( I_3 + I_4 + 2 I_5 \right). \tag{A10}$$

Terms $I_3$ and $I_4$ are collectively defined as:

$$I_3 + I_4 = \int_0^\infty \left( e^{2D_+} + e^{2D_-} \right) r \, \mathrm{d}r = \lim_{R \to \infty} \int_0^R r e^{\frac{-(r+r_0)^2}{\sigma^2}} + r e^{\frac{-(r-r_0)^2}{\sigma^2}} \, \mathrm{d}r. \tag{A11}$$

Solving the integral yields:

$$I_3 + I_4 = \lim_{R \to \infty} \left[ \frac{-\sigma^2}{2} \left( e^{\frac{-(r+r_0)^2}{\sigma^2}} + e^{\frac{-(r-r_0)^2}{\sigma^2}} \right) - \frac{\sqrt{\pi}}{2} r_0 \sigma \left( \mathrm{erf}\left( \frac{r+r_0}{\sigma} \right) - \mathrm{erf}\left( \frac{r-r_0}{\sigma} \right) \right) \right]_0^R , \tag{A12a}$$

$$= \sigma^2 e^{\frac{-r_0^2}{\sigma^2}} + \sqrt{\pi} r_0 \sigma \, \mathrm{erf}\left( \frac{r_0}{\sigma} \right). \tag{A12b}$$



Finally, $I_5$ is defined as:

$$I_5 = \int_0^\infty e^{D^+ + D^-} r \, dr = \lim_{R \to \infty} \int_0^R r e^{\left( \frac{-(r+r_0)^2}{2\sigma^2} - \frac{(r-r_0)^2}{2\sigma^2} \right)} dr, \tag{A13}$$

which, once integrated, gives

$$I_5 = \lim_{R \to \infty} \left[ \frac{-\sigma^2}{2} e^{\frac{-\left(r^2 + r_0^2\right)}{\sigma^2}} \right]_0^R, \tag{A14a}$$

$$= \frac{\sigma^2}{2} e^{\frac{-r_0^2}{\sigma^2}}. \tag{A14b}$$

Therefore, one gets

$$N = \frac{1}{2} \left( I_3 + I_4 + 2I_5 \right) = \sigma^2 e^{\frac{-r_0^2}{\sigma^2}} + \frac{\sqrt{\pi}}{2} r_0 \sigma \operatorname{erf}\left( \frac{r_0}{\sigma} \right), \tag{A15}$$

which corresponds to Eq. (7b).

*Competing interests.* The authors declare that they have no conflict of interest.

*Acknowledgements.* This work has been partially supported by the CL-Windcon project, which receives funding from the European Union
Horizon 2020 research and innovation programme under grant agreement No. 727477.



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
