# Peer review of "Brief communication: A double Gaussian wake model"

_Wind Energy Science, 2019_

## Referee Comment (RC1) · Anonymous Referee #1 · 12 Sep 2019

This is an interesting paper with some developments on the double Gaussian wind turbine wake model.

The model is validated with experimental measurements obtained with scaled wind turbine models and corresponding large eddy simulations.

The parameter kr, representing the position of the gaussian extrema, is not fixed as in the original model, but is tuned to facilitate a suitable fit to the data. In addition, the wake expansion function sigma(x) is taken to be linear in the downstream distance x, as proposed by Bastankhah and Porte-Agel (2014).

The results show good agreement with the reference data, in particular the double Gaussian model showing better agreement than the single Gaussian model in the near

wake.

---

## Referee Comment (RC2) · M. J. Churchfield (Referee) · 11 Nov 2019

This is a manuscript for a short article describing a new engineering wake model that uses a double-Gaussian formulation rather than the more common single-Gaussian formulation. This is a useful addition to wake modeling because wake deficit profiles appear double-Gaussian in the near wake. Such a model as this increases the range of distances over which the wake model can be accurate. Often, proposed wake models include text stating that they are not designed to be accurate in the near wake.

The manuscript is well written. I enjoyed reading it. I have some minor comments outlined below that are meant to strengthen the paper. I would like the authors to consider these comments, and I recommend this manuscript for publication.

**Minor Items:**

- Page 1, line 25: Is the statement "Nowadays, onshore wind farms tend to be closely packed, and turbine spacing often reaches values below 3D," really true? I realize you are using this for motivation of having accurate near-wake behavior, but I cannot think of a wind farm with such tight spacing. Lillgrund is a tightly packed farm, and it is just above 3D in one of the directions.

- Page 2, line 31: You say "This is due to the presence of two peaks in the speed profiles close to the rotor disk." Please be precise with how you word this. Make sure to say that they are **peaks** in the **velocity deficit profiles** or **minima** in the **velocity profiles**.

- Page 3, equation 3: I find it confusing to label momentum as $p$ when there is pressure in the full momentum equation and also we speak of power a lot with wind turbines.

- Page 6, section 3.1: Just to be clear, there is no vertical shear in the inflow wind profile, correct? Therefore, $U_\infty$ in your equations is just a single constant–it is not dependent on $y$, right?

- Page 7, line 144: The word "aeroservoelastic" is misspelled.

- Page 7, line 145: I think you mean to say "non-uniformity", and not "non-uniformly."

- Page 9, figure 4: The single Gaussian and double Gaussian RMSE become nearly equal at 9D. You should comment on what you think may happen further downstream beyond 9D. Do you think that because the double Gaussian model is at its core two Gaussians, it would do a worse job than the single Gaussian far downstream where real data appears more single Gaussian, or do you think the

blend of the two Gaussians is sufficient? I ask because although typical larger spacings are around $9D$, the wake of the most upstream turbine continues on past the second row to the third, fouth, and so on.

- General comment: I like this idea of the double Gaussian very much. In our LES of turbines in sheared inflow, we notice that both the spanwise and vertical wake profiles are double Gaussian, but where the amplitude of each Gaussian is different. This makes sense vertically because of the vertical shear. Horizontally, we conjecture that the vertical asymmetry gets rotated to a horizontal asymmetry due to the wake rotation in the near wake. Would you ever incorporate something like this into your model? This seems important if people are to use it to predict wakes in real sheared atmospheric flow.

- General comment: I know this is just a short paper, but do you have plans to tune this to more data to come up with a constant $k^*$ that is dependent on background turbulence intensity? Also, the location of the peaks of the Gaussians relative to centerline seems very dependent on the loading profile of the rotor. Can you somehow fit to enough data to make $k_r$ tuning-free for the user?

---

## Author Comment (AC1) · 20 Nov 2019

Please see attached letter with detailed reply to both reviewers, and a revised manuscript with changes highlighted in red.

Please also note the supplement to this comment:
https://www.wind-energ-sci-discuss.net/wes-2019-52/wes-2019-52-AC1-supplement.pdf

---

## Author Response (AR1)

**Reply to Reviewers**

We thank the reviewers for their constructive inputs. Where applicable, point-by-point replies to their comments are detailed in the following.

**Reviewer 1**

**Reviewer**:
*"This is an interesting paper with some developments on the double Gaussian wind turbine wake model. The model is validated with experimental measurements obtained with scaled wind turbine models and corresponding large eddy simulations.*
*The parameter kr, representing the position of the gaussian extrema, is not fixed as in the original model, but is tuned to facilitate a suitable fit to the data. In addition, the wake expansion function sigma(x) is taken to be linear in the downstream distance x, as proposed by Bastankhah and Porte-Agel (2014).*
*The results show good agreement with the reference data, in particular the double Gaussian model showing better agreement than the single Gaussian model in the near wake."*
**Authors**: Thank you for your comments.

**Reviewer 2**

**Reviewer**:
*"This is a manuscript for a short article describing a new engineering wake model that uses a double-Gaussian formulation rather than the more common single-Gaussian formulation. This is a useful addition to wake modeling because wake deficit profiles appear double-Gaussian in the near wake. Such a model as this increases the range of distances over which the wake model can be accurate. Often, proposed wake models include text stating that they are not designed to be accurate in the near wake.*

*The manuscript is well written. I enjoyed reading it. I have some minor comments outlined below that are meant to strengthen the paper. I would like the authors to consider these comments, and I recommend this manuscript for publication."*

**Authors**: Thank you for your words. A point-by-point reply to your comments is reported here below.

1. **Reviewer**: *Page 1, line 25: Is the statement "Nowadays, onshore wind farms tend to be closely packed, and turbine spacing often reaches values below 3D," really true? I realize you are using this for motivation of having accurate near-wake behavior, but I cannot think of a wind farm with such tight spacing. Lillgrund is a tightly packed farm, and it is just above 3D in one of the directions.*
   **Authors**: There are indeed wind farms with such tight spacing: the onshore wind farm investigated in ref. [1] shows distances of up to 3.0 D in the sub-cluster reported in Fig. 4. From Fig. 1 of the same publication, it appears that even closer turbine spacing are can be found within that wind farm. In addition, ref. [2] reports that an onshore wind farm in Germany has spacings in the

prevailing wind direction of below 3.5 D and in the orthogonal direction of below 3 D. The minimum distance in that example is as low as 2.2 D.

Note that onshore wind farms might have smaller turbine spacing than offshore farms (Lillgrund).

We included the following two references in the revised version of the manuscript and rephrased slightly the sentence to "turbine spacing often reaches values close to or even below 3D."

[1] Schreiber, J., Salbert, B., and Bottasso, C. L.: Study of wind farm control potential based on SCADA data, Journal of Physics: Conference Series, 1037, 032 012, https://doi.org/10.1088/1742-6596/1037/3/032012, 2018
[2] energiespektrum.de: Produzieren auf engem Raum, https://www.energiespektrum.de/produzieren-auf-engem-raum-8918, 2015, accessed: 2019-11-12.

2. **Reviewer**: *Page 2, line 31: You say "This is due to the presence of two peaks in the speed profiles close to the rotor disk." Please be precise with how you word this. Make sure to say that they are peaks in the velocity deficit profiles or minima in the velocity profiles.*
   **Authors**: We rephrased the sentence to "This is due to the presence of two minima in the speed profiles close to the rotor disk".

3. **Reviewer**: *Page 3, equation 3: I find it confusing to label momentum as p when there is pressure in the full momentum equation and also we speak of power a lot with wind turbines.*
   **Authors**: The symbol p is commonly used in the literature to indicate momentum. The authors believe that changing it to another symbol might cause confusion to the reader. Note that pressure and power, which might also be noted by the symbol p (although the latter customarily uses P instead of p), are not used in the present paper, so that no confusion should arise here.

4. **Reviewer**: *Page 6, section 3.1: Just to be clear, there is no vertical shear in the inflow wind profile, correct? Therefore, $U_\infty$ in your equations is just a single constant–it is not dependent on y, right?*
   **Authors**: The experiments have been conducted with a vertically sheared inflow denoted by a power law exponent equal to 0.144 (now included in the revised text). The wake model is formulated under the assumption of uniform inflow, therefore $U_\infty$ is indeed a single constant in the equations. When applying the wake model, $U_\infty$ has been chosen as the velocity at hub height, which is also the height at which the wake measurements are taken.

5. **Reviewer**: *Page 7, line 144: The word "aeroservoelastic" is misspelled.*
   **Authors**: This has now been fixed.

6. **Reviewer**: *Page 7, line 145: I think you mean to say "non-uniformity", and not "nonuniformly."*
   **Authors**: This has now been fixed.

7. **Reviewer**: *Page 9, figure 4: The single Gaussian and double Gaussian RMSE become nearly equal at 9D. You should comment on what you think may happen further downstream beyond 9D. Do you think that because the double Gaussian model is at its core two Gaussians, it would do a worse job than the single Gaussian far downstream where real data appears more single Gaussian, or do*

*you think the blend of the two Gaussians is sufficient? I ask because although typical larger spacings are around 9D, the wake of the most upstream turbine continues on past the second row to the third, fouth, and so on.*

**Authors**: This point is addressed by Fig. 4, which shows that the two parameterizations are nearly identical in the far wake. To make this point clearer, we have added the following text to the manuscript: "Comparing the SG with the DG model, Fig. 4 shows that both reach very similar RMSEs for 9 D. The similarity between the two models continues for larger downstream distances." In addition, we have added the following suggestion for future work in the conclusions: "The different shape of the wake in the near and far regions could also suggest two stitch together the two models, the double Gaussian being used in the near wake and single Gaussian further downstream, thus avoiding the need for a single tuning that has to cover such a long distance and different behaviors."

8. **Reviewer**: *General comment: I like this idea of the double Gaussian very much. In our LES of turbines in sheared inflow, we notice that both the* spanwise *and vertical wake profiles are double Gaussian, but where the amplitude of each Gaussian is different. This makes sense vertically because of the vertical shear. Horizontally, we conjecture that the vertical asymmetry gets rotated to a horizontal asymmetry due to the wake rotation in the near wake. Would you ever incorporate something like this into your model? This seems important if people are to use it to predict wakes in real sheared atmospheric flow.*

   **Authors**: As already mentioned, the inflow within the experiments is also vertically sheared. A small asymmetry of the peaks can indeed be seen in measurements and CFD. A nonsymmetric double Gaussian could also be observed in wind-misaligned conditions, as already noted at the very end of the conclusions section. These are interesting ideas, which however have not been studied enough by these authors to be included in the present paper. In any case, we have added the following sentence to the revised manuscript: "A possible improvement to the present model might include an azimuth-dependent double Gaussian function. This would allow one to model a non-axialsymmetric double-peaked wake profile, caused by a sheared inflow and/or by the misalignment of the rotor axis with respect to the wind, at the cost of extra tuning parameters." Also note that, as mentioned in the paper, the double Gaussian model has been first published by Keane et al. The goal of our paper is mainly to correct some mistakes in the original formulation.

9. **Reviewer**: *General comment: I know this is just a short paper, but do you have plans to tune this to more data to come up with a constant $k_*$ that is dependent on background turbulence intensity? Also, the location of the peaks of the Gaussians relative to centerline seems very dependent on the loading profile of the rotor. Can you somehow fit to enough data to make $k_r$ tuning-free for the user?*

   **Authors**: Certainly, a turbulence dependent baseline parameterization, ideally using full scale measurements, would be of great value for any user of the model. We think that this would be a very useful development, which however falls outside of the scope of the present work.

A revised version of the manuscript is attached to the present reply, with all changes highlighted in red.

Best regards.
The authors

[revised manuscript text omitted]

---

## Author Response (AR2)

Dear Associate Editor,

Thank you for your message.

You write: *you have still not addressed my issue about the use of "ansatz velocity deficit distribution".*

Indeed, we already replied on 23 August 2019, and you can find our letter in the MS Records on the journal site. We copy here below again our original reply.

> Dear Associate Editor,
>
> Thank you for your initial assessment of our paper.
>
> The word "ansatz" is not *"a relic of German language left from an earlier version"*. Indeed, ansatz is a common term used in mathematical language to refer to an "assumption". A google search returns over 34 million results, and the definition "an assumption about the form of an unknown function which is made in order to facilitate solution of an equation or other problem". Wikipedia, citing Gershenfeld 1999, states that ansatz "is an educated guess that is verified later by its results".
>
> This is exactly the use that we make of this word in our paper: we assume (based on prior knowledge) that a double Gaussian is a good choice for the shape function describing the velocity deficit in a wind turbine wake, and then we verify a posteriori that is indeed a better choice than others.
>
> Thank you again for your help in reviewing this paper.
>
> The authors
>
> References
> Gershenfeld, Neil A. (1999), The Nature of Mathematical Modeling, Cambridge University Press, ISBN 0-521-57095-6

Because ansatz is a common term used in technical publications, we prefer not to change the manuscript.

Thank you again for your help in reviewing this paper

The authors